## [Peer Review file · Nature]

Genomic history of early dogs in Europe

Corresponding Author: Dr Anders Bergström

Version 1:

Reviewer comments:

Referee #1

(Remarks to the Author)

The origin and early evolution of domestic dogs is equally captivating and illusive, but has major implications for our understanding of human history, domestication and evolutionary biology overall.

In this regard, the work presented by Bergström et al. represent an substantial advancement of the field, especially in Europe but with important perspectives on global dogs. By generating and analysing genome-wide data from 215 ancient canid specimens, most from pre-Neolithic Europe, the authors provide unprecedented population genomic resolution into the history of early European dogs, their relationship with wolves, correlated with major cultural transitions, such as the advent of farming.

Among the most striking findings is the recovery of genome-wide data from a ~14200 year old dog from Switzerland. This specimen shares substantial ancestry with later European dogs, indicating that diversification within the domestic dog lineage was well underway by the Upper Palaeolithic. This pushes back the timeline for dog domestication and early population structure beyond what has previously been demonstrated. Most interestingly, this Swiss Palaeolithic dog is noteworthy different to Mesolithic Northern European dogs, indicating that more findings are yet to be done in regards to dog history, even within Europe. But importantly, demonstrating that the findings and data made here is setting the path for future research. The authors also assess the demographic impact of Neolithic human migrations into Europe, interestingly and unlike human populations, dogs experienced only partial genetic turnover. Contrasting dog demographics during other comprehensive historical transitions.

The solid findings and impressive approaches of the study are achievements on their own, but also opens up exciting new avenues for future research on the domestication process and reinforces the utility of ancient genomic data in resolving deep evolutionary questions.

This is an exceptional and rudimentary contribution, likely to serve as a key reference point in dog genomics and the study of dog domestication.

I have a few comments and suggestions for the authors, after which I would be happy to see this work published in Nature.

Suggested improvements:

I found the total number of samples processed and total number of samples yielding data included in analysis a bit confusing.

From the text:

Line 88 "We analysed the genomes of 215 canid remains"

Line 135 "We extracted and analysed DNA from 215 canid skeletal remains"

Line 175-176 "We successfully obtained this resolution and a dog/wolf identification for 142 out of the 215 analysed remains (134 out of 138 for samples subjected to capture)."

Then the table "Ancient genome metadata" list 219 samples, 6 of which has 0 capture sites covered and NA shotgun coverage. Further, 40 samples has NA capture sites covered and NA shotgun coverage.

Please consider clarifying the numbers a bit, I think it would be useful to state clearly how many samples were processed

and how many are included in data analysis

Until the very end of the method section, I was confused about the oldest dog/dogs in the study, you clearly write many times, the oldest dog is the 14.2k year-old Kesslerloch specimen. However, 4 samples are directly dated older as part of the study and found to be dogs, specifically;

TU1078 Gnrishohle Gnrishöhle GN1; E/52-106 15566 Cal BP mean

TU1077 Gnrishohle Gnrishöhle GN1; F/52-14 15506 Cal BP mean

GNH005 Gnrishohle Gnrishöhle GN1; F/50-70 15323 Cal BP mean

GNH006 Gnrishohle Gnrishöhle GN1; E/52-114 15404 Cal BP mean

Yet, in the supplementary overview these have an “Analysis date” of 100, based genetic inference.

The main text touch upon this matter;

“radiocarbon dating or analysis of the genetic data demonstrates that these dogs must have lived in the last few thousand years, meaning that these findings must reflect either later deposition of dog bones at these sites, or DNA contamination from more recent dogs”

However given the value of potential dog data of this age I do not find the explanation for conflicting genetic inference and C14 dating sufficient.

Finally in the end of the methods it is explained and well supported that these samples are contaminated. Consider clarifying this in the main text, you clearly put some work I to this, so it is worth underlining you find clear signs of contamination, unfortunately.

Are these (all dubious) samples removed from analyses/figures?

Minor comments:

It might be helpful to clarify somewhere that “ka” refers to 'thousand years ago,' just to ensure it's clear for all readers. Same for “ky.

I noticed that sometimes you use “Fig” and other times “Fig.”. You might want to consider standardising this for consistency throughout the manuscript.

In section “Early dogs in Europe share origins with other worldwide dogs”, I think you refer to Fig 3a-b instead of Fig 2a-b?

Line “144”

“Ertebølle”, this is the only example of this spelling, throughout the text Ertebølle is used.

179-181

“we identify 21 dogs and four wolves, the latter being from the Ertebølle and Ramløse sites and previously identified as wolves morphologically.”

Is there a reference for the wolf identification? Or could a statement be added like “according to museum records?”

Line 330-332

“The Anatolian Boncuklu dog (11.4 ka) displays a similarly divergent profile, though less pronounced than in Kesslerloch (Extended Data Figure 4a).”

I am sorry I don't understand how this is shown in Extended Data Figure 4a. Sorry if I am misunderstanding the figure and in that case just ignore my comment.

Line 765-768

“Principal components analysis was performed using EIGENSOFT v7.2.173, using the “poplistname” parameter to specify 211 modern Eurasian wolves to use for inferring principal components, with dogs instead being projected.”

Could you add an overview of the samples used in this PCA?

There is only 140 wolves in the supplementary overview...

Figure 3b

Should the age of Zhokhov be 9,5k?

In supplementary table “Previously published genomes”

Is there no publication/citation for 722g? or some more meaningful reference?

In supplementary table “Previously published genomes”

I think “Wolf08Unknown” is BioSample: SAMN03801694; Sample name: Wolf08; SRA: SRS984773

In supplementary table “Previously published genomes”

The samples;

Coyote_Mexico CoyoteMexico SAMN10180422

GreyWolf_Daneborg WolfDaneborg SAMN10180430

Are listed as published in Sinding et al. 2018, I believe they were released in Gopalakrishnan et al. 2018

Altogether a thorough and valuable study.

Best,

Mikkel Sinding

Referee #2

(Remarks to the Author)

The manuscript "Genomic history of early dogs in Europe" by Bergström et al. represents a significant advancement in our understanding of early dog domestication in Europe, as informed by ancient DNA analysis. The authors present a remarkably large dataset comprising 215 canid specimens, 178 of which are pre-Neolithic, making this one of the most comprehensive genomic studies of ancient dogs to date. The use of targeted enrichment to recover genomic data from Upper Paleolithic and Mesolithic samples with low endogenous content represents an important methodological contribution, setting a benchmark for future research. The study offers valuable insights into the origins and diversification of domestic dogs, their relationships with wolves, and the demographic impacts of the Neolithic transition. These findings significantly advance our understanding of domestication processes and the long-term co-evolution of dogs and humans. Nonetheless, several aspects of the manuscript could be further refined to strengthen its overall impact.

1. For the qpAdm analyses, please provide more details on the choice of reference populations as well as the outgroups. Sensitivity analyses (e.g., varying reference sets) would strengthen the conclusions about genetic ancestry and admixture proportions.
2. The study covers a broad time period (14–4 ka), but some regions have denser sampling than others. Acknowledge potential biases due to uneven sampling and discuss how this might affect interpretations.
3. I would suggest using chromosome painting (e.g., with GLOBETROTTER) to estimate admixture timing and sources more precisely.
4. Since ADMIXTURE is extensively described in this study, please assess stability by running multiple replicates with different random seeds.
5. This study obtained time-series samples from Europe, which would be important to estimate effective population size change across time, especially pre- and post-Neolithic period.
6. Even though nuclear data has higher resolution, uniparental genetic markers should also be mentioned in the main text.

Some minor errors that I noticed:

1. References 8 and 20 both cite Frantz et al. (2016) with identical titles but different links.
2. Correct spelling errors (e.g., "Anotalian" to "Anatolian", line 260)
3. Check Fig. 2a, the zoomed-in PCA, whether pre-agriculture and post-agriculture are correctly labeled.

Referee #3

(Remarks to the Author)

This is a good paper, with very interesting results about the complexity of dog ancestry and its relationship to past population movements and economic changes.

I am an archaeologist, not a geneticist. With that caveat, I do not have major concerns. My main technical comment is that I would like to see more details about the radiocarbon dating and the potential (or lack thereof) for radiocarbon reservoir offset on the directly dated specimens (see my more specific comments below).

Besides that, I would recommend additional contextualizing of some of the main implications of these findings. In several instances, the authors link the patterns they observe (on dogs) to known archaeological patterns about people, the most striking example being the increase in near-eastern ancestry in Europe following the Neolithic. The authors are quite terse on the implications of these findings, and what social or economic processes, intentional or not, may have been at play. What does this finding imply about social relationships between human groups, and the roles that dogs may have played? I would also suggest being more explicit about some reasoning shortcuts. I list a few below; the authors for instance implicitly assume that all ancient dogs must be genetically related to modern dogs; or that a sampled bone sort of equates an ancient individual.

I list some more comments below.

line 106. indicate that (presumably) all dates in this paper are calibrated

line 108: reference 17 for the Americas seems a bit outdated – surely there is more recent work to be cited here, if nothing else by some of the authors themselves

lines 135-144. It would be good to explain how samples were selected. Do they represent different individuals, and if so how did the authors make sure they do? I imagine for instance that the 103 samples from Kesslerloch do not represent a MNI of 103. If they represent the same individuals, how does it affect the results?

lines 171-180. the authors start by discussing how their test separates "known dogs" from wolves, then switch to how it separates [implicitly all] "dogs" from wolves. Perhaps they ought to discuss how they make this leap, or at least explicitly state it. Is it inconceivable that there could be ghost dog lineages unrelated to "known dogs"? If it is so, could the authors explain why?

lines 178-188. can the authors indicate the time frame for all sites? as it stands, it is only given for a select few. This

comment applies to the entire article.

line 187-188. the statement on hunting being the most parsimonious explanation comes a bit out of nowhere. I would suggest either expanding on this argument or removing it.

line 192. does this finding contradict the argument above that wolves were obtained by hunting?

line 194: other ways such as?

lines 198-201. are the bones found in stratigraphic contexts that suggest a late deposition?

line 202-204. likewise, can the authors explain this major dating discrepancy?

line 206-207. this partially answers my query for lines 135-144, but only partially. does this mean Kesslerloch has 62 individuals instead of 103? What of the other sites?

line 207. is the dog from the same archaeological context than the wolves? it seems odd to be by itself among so many wolves.

lines 170-211. Is there any dietary information on these dogs, such as stable isotopes? Given the likelihood that at least some of these people (and their dogs) may have eaten aquatic resources, the authors should discuss how radiocarbon reservoir effect may affect their dates (or not).

lines 343-346. can the authors relate this SW Asia / Magdalenian canid connection with connections (or lack thereof) observed among human populations at that time?

line 412-414. this statement makes it sound as if "Mesolithic people" were replaced by "Neolithic people" and went extinct. Many of these farmer groups probably included people (and their dogs) from Mesolithic backgrounds.

line 445-446. I see that this responds to my comment in lines 343-346. It could be good to mention that earlier.

lines 455-456. here (or somewhere else) the authors could cite the new paper by Manin et al. that discusses how dog populations were affected by the "Neolithic transition" in South America.

lines 455-456. It could also be good to discuss the differing roles that dogs may have played among hunter-gatherers vs farmers populations.

lines 582-584. As mentioned in a previous comment, it would be good to explain the discrepancy here.

lines 579-584. As mentioned earlier, the authors should discuss the impact (or lack thereof) of diet on potential radiocarbon reservoir effect.

Supplemental excel file. The large variation in $\delta^{13}C$ from -14 to -24 makes me wonder, again, how diet may affect radiocarbon dates (particularly the values above -14). Are those values expected of canids with fully terrestrial diets for these areas? Are $\delta^{15}N$ available? Are isotopic values ($\delta^{13}C$ and $\delta^{15}N$) available for the dated Kesslerloch specimens?

François Lanoë

Version 2:

Reviewer comments:

Referee #1

(Remarks to the Author)

I am delighted to say that after reviewing the revised manuscript and the authors detailed responses to my previous comments. All points have been thoroughly and thoughtfully addressed, and the revisions have clearly strengthened the paper. I am fully satisfied with the changes and have no further comments or suggestions.

Best,
Mikkel Sinding

Referee #2

(Remarks to the Author)

The revision for Bergström et al., "Genomic history of early dogs in Europe" in my view has addressed the key methodological and interpretative concerns raised in the previous review round. The logic of the population-genetic framework is now consistent throughout, and the manuscript reads clearly and coherently. I have no reservations seeing it published in this form. I only have a few very minor editorial suggestions for clarification and consistency:

1. Please replace "two-thirds of the tested remains" in the Abstract with "and were able to determine dog-wolf ancestry for 142 of 216 tested remains," as giving the explicit number improves clarity and allows readers to compare it with the dataset description in the Methods directly.

2. The comparison between human and dog demographic histories during the Neolithic transition is well motivated, but could be made more cohesive. In particular, the authors note that Southwest Asian ancestry contributed less to dogs than to humans, and that dogs do not show the increase in heterozygosity seen in human Neolithic populations (Fig. 2c). I would suggest explicitly linking these two observations in the discussion. Doing so would emphasize that the difference between humans and dogs reflects not only a smaller ancestry replacement but also a weaker demographic expansion in dogs, thereby providing a more integrated picture of how domesticated and human populations responded differently to the Neolithic transition.

Referee #3

(Remarks to the Author)

The authors answered all my queries appropriately, and I appreciate their effort. I am looking forward to seeing this article published.

-François Lanoë

We thank all three reviewers for their comments, and are pleased that they have evaluated our study positively. We respond to their individual comments and suggestions in what follows.

Additionally, in response to various reviewer questions surrounding sample information, archaeological context, metadata etc. we have added the following to the paper, to provide more detailed information to readers about the samples and datasets reported:

- We have added detailed archaeological descriptions of all sites and samples studied as supplementary material (Supplementary Notes).
- We have added a supplementary table containing information on what sampled skeletal remains could potentially come from the same individual, based on prior zooarchaeological assessments (Supplementary Data 5)
- To ensure reproducibility and maximise utility of our capture approach to other researchers, we have deposited files describing the targeted SNPs and the sequences of the DNA probes used to Zenodo, and include the DOI in the Data Availability Statement
- As further characterisation of the generated genomic data, we have added a Extended Data Table describing possible pathogens detected in the data (Extended Data Table 1)

Referee #1 (Remarks to the Author):

The origin and early evolution of domestic dogs is equally captivating and illusive, but has major implications for our understanding of human history, domestication and evolutionary biology overall.

In this regard, the work presented by Bergström et al. represent an substantial advancement of the field, especially in Europe but with important perspectives on global dogs. By generating and analysing genome-wide data from 215 ancient canid specimens, most from pre-Neolithic Europe, the authors provide unprecedented population genomic resolution into the history of early European dogs, their relationship with wolves, correlated with major cultural transitions, such as the advent of farming.

Among the most striking findings is the recovery of genome-wide data from a ~14200 year old dog from Switzerland. This specimen shares substantial ancestry with later European dogs, indicating that diversification within the domestic dog lineage was well underway by the Upper Palaeolithic. This pushes back the timeline for dog domestication and early population structure beyond what has previously been demonstrated. Most interestingly, this Swiss Palaeolithic dog is noteworthy different to Mesolithic Northern European dogs, indicating that more findings are yet to be done in regards to dog history, even within Europe. But importantly, demonstrating that the findings and data made here is setting the path for future research. The authors also assess the demographic impact of Neolithic human migrations into Europe, interestingly and unlike human populations, dogs experienced only partial genetic turnover. Contrasting dog demographics during other comprehensive historical transitions.

The solid findings and impressive approaches of the study are achievements on their own, but also opens up exciting new avenues for future research on the domestication process and reinforces the utility of ancient genomic data in resolving deep evolutionary questions. This is an exceptional and rudimentary contribution, likely to serve as a key reference point in dog genomics and the study of dog domestication.

I have a few comments and suggestions for the authors, after which I would be happy to see this work published in Nature.

We thank the reviewer for their kind words and very helpful suggestions, including catching several hard-to-spot mistakes in the details of the supplementary materials and metadata, which we have thus been able to correct.

Suggested improvements:

I found the total number of samples processed and total number of samples yielding data included in analysis a bit confusing.

From the text:

Line 88 “We analysed the genomes of 215 canid remains”

Line 135 “We extracted and analysed DNA from 215 canid skeletal remains”

Line 175-176 “We successfully obtained this resolution and a dog/wolf identification for 142 out of the 215 analysed remains (134 out of 138 for samples subjected to capture).”

Then the table “Ancient genome metadata” list 219 samples, 6 of which has 0 capture sites covered and NA shotgun coverage. Further, 40 samples has NA capture sites covered and NA shotgun coverage.

Please consider clarifying the numbers a bit, I think it would be useful to state clearly how many samples were processed and how many are included in data analysis

We agree and understand that this is somewhat confusing, due to the large number of samples included in the study and the different types of data generated for different subsets of samples.

Firstly, one potential source of confusion is that the number of sampled remains does not necessarily correspond to the number of unique biological individuals. We have removed a few confirmed instances of multiple samples from the same individual from the Supplementary table, such that the number of rows in that table now matches the 216 cited in the main text. To clarify more generally, we have added the following paragraph to the beginning of the Methods section describing the sampling (as well as a few other changes described in response to reviewer 3):

“When sampling skeletal remains for DNA, in most instances we aimed to take one sample per biological individual. In some instances, samples were obtained from different remains that possibly could derive from the same individual, based on the zooarchaeological context (in particular this was the case for the Kesslerloch site). In a few cases where it was unambiguous that data from multiple remains came from the same individual, we merged those data. But in most cases, we treated data from different remains separately, to err on the side of caution and not incorrectly merge data. The true number of sampled individuals is thus unknown, but we provide a table listing the groups of remains where archaeological information suggest they might derive from the same individual (Supplementary Data 5).”

Secondly, in terms of the sequencing performed, different samples have different combinations of low-coverage screening, capture, and deep shotgun sequencing. In the supplementary table, the three columns “Shotgun”, “Capture”, and “Screening” use 1 and 0 to denote whether a given data type was generated for a given sample - we have amended these column names to “Shotgun data generated” etc. to make it clearer that this is what they represent. The column “Capture sites covered” can have a non-zero value even if a sample was not captured, if the screening or shotgun data covered any of those sites (whereas an NA values in this column means the data was so poor we did not count how many sites were covered).

Until the very end of the method section, I was confused about the oldest dog/dogs in the study, you clearly write many times, the oldest dog is the 14.2k year-old Kesslerloch specimen. However, 4 samples are directly dated older as part of the study and found to be dogs, specifically;

TU1078 Gnirshohle Gnirshöhle GN1; E/52-106 15566 Cal BP mean

TU1077 Gnirshohle Gnirshöhle GN1; F/52-14 15506 Cal BP mean

GNH005 Gnirshohle Gnirshöhle GN1; F/50-70 15323 Cal BP mean

GNH006 Gnirshohle Gnirshöhle GN1; E/52-114 15404 Cal BP mean

Yet, in the supplementary overview these have an a “Analysis date” of 100, based genetic inference.

The main text touch upon this matter;

“radiocarbon dating or analysis of the genetic data demonstrates that these dogs must have lived in the last few thousand years, meaning that these findings must reflect either later deposition of dog bones at these sites, or DNA contamination from more recent dogs”

However given the value of potential dog data of this age I do not find the explanation for conflicting genetic inference and C14 dating sufficient.

Finally in the end of the methods it is explained and well supported that these samples are contaminated. Consider clarifying this in the main text, you clearly put some work I to this, so it is worth underlining you find clear signs of contamination, unfortunately.

Are these (all dubious) samples removed from analyses/figures?

We see how the contaminated Gnirshöhle samples could be a source of confusion for readers, even if the details are explained in the Methods. To help prevent confusion, we have now removed them from main text Fig 1C (we still keep them in the ADMIXTURE plot in Figure 3, to allow readers to inspect their ancestry profiles). We have also removed the potentially confusing age of 100 for their Analysis dates in the metadata table, instead indicating the age of the actual specimens. We have also changed their values in the “Dog or wolf ancestry” column in the metadata table from “Dog” to “Dog (modern contamination)”, to very clearly flag them. Finally, in the sentence introducing them in the main text, we have changed from “we identified eight dogs” to “we identified eight remains displaying dog ancestry”, and removed the phrasing “these dogs must have lived in the last few thousand years, so as to not refer to these samples as dogs. Hopefully these changes should go a long way towards preventing confusion.

One possibility would have been to completely exclude these contaminated samples from the paper and not even mention them, but we believe it is better to transparently report everything we find - not least in order to prevent confusion if someone else wanted to revisit

these samples in the future. We also believe including these results more accurately represents the actual scientific process: if you go out and look for the earliest dogs in Europe, modern dog contamination of old wolf bones is an issue that you might run into, and need to deal with.

Minor comments:

It might be helpful to clarify somewhere that “ka” refers to 'thousand years ago,' just to ensure it's clear for all readers. Same for “ky.

We now write “thousand years ago (ka)” at first usage, in the abstract. To simplify things, we have removed usage of “ky” and instead spell it out in the few instances where this was used.

I noticed that sometimes you use “Fig” and other times “Fig.”. You might want to consider standardising this for consistency throughout the manuscript.

We have standardised all instances to “Fig.”.

In section “Early dogs in Europe share origins with other worldwide dogs”, I think you refer to Fig 3a-b instead of Fig 2a-b?

We have corrected this and now refer to Fig. 2 in this section.

Line “144”

“Ertebølle”, this is the only example of this spelling, throughout the text Ertebølle is used.

We have corrected this to “Ertebølle”.

179-181

“we identify 21 dogs and four wolves, the latter being from the Ertebølle and Ramløse sites and previously identified as wolves morphologically.”

Is there a reference for the wolf identification? Or could a statement be added like “according to museum records?”

To our knowledge there is no published reference for these morphology assignments, and so we have followed the reviewers suggestion and added “according to museum records”.

Line 330-332

“The Anatolian Boncuklu dog (11.4 ka) displays a similarly divergent profile, though less pronounced than in Kesslerloch (Extended Data Figure 4a).”

I am sorry I don't understand how this is shown in Extended Data Figure 4a. Sorry if I am misunderstanding the figure and in that case just ignore my comment.

Extended Data Fig. 4a is the same plot as the main Fig. 3C, but using the Anatolian Boncuklu dog in place of Kesslerloch on the horizontal axis. Fig. 3C shows that almost all dogs are closer to the Scandinavian dog than to Kesslerloch, as almost all dogs are above the diagonal line. In a similar fashion, Extended Data Fig. 4a shows that almost all dogs are

close to the Scandinavian dog than to Boncuklu, as here too they are mostly above the diagonal line.

Line 765-768

“Principal components analysis was performed using EIGENSOFT v7.2.173, using the “poplistname” parameter to specify 211 modern Eurasian wolves to use for inferring principal components, with dogs instead being projected.”

Could you add an overview of the samples used in this PCA?

There is only 140 wolves in the supplementary overview...

This is a good point, and we have now added a table listing the wolf samples used in the PCA as Supplementary Data 4.

Figure 3b

Should the age of Zhokhov be 9,5k?

We have corrected this to 9.5k.

In supplementary table “Previously published genomes”

Is there no publication/citation for 722g? or some more meaningful reference?

We have replaced “722g” with “Plassais et al. 2019”, which is the paper describing the 722g resource.

In supplementary table “Previously published genomes”

I think “Wolf08Unknown” is BioSample: SAMN03801694; Sample name: Wolf08; SRA: SRS984773

We thank the reviewer for this pointer and have added in the BioSample ID.

In supplementary table “Previously published genomes”

The samples;

Coyote_Mexico CoyoteMexico SAMN10180422

GreyWolf_Daneborg WolfDaneborg SAMN10180430

Are listed as published in Sinding et al. 2018, I believe they were released in Gopalakrishnan et al. 2018

We thank the reviewer for noticing this, and we have corrected the citations for these to Gopalakrishnan et al. 2018.

Altogether a thorough and valuable study.

Best,

Mikkel Sinding

Referee #2 (Remarks to the Author):

The manuscript "Genomic history of early dogs in Europe" by Bergström et al. represents a significant advancement in our understanding of early dog domestication in Europe, as informed by ancient DNA analysis. The authors present a remarkably large dataset comprising 215 canid specimens, 178 of which are pre-Neolithic, making this one of the most comprehensive genomic studies of ancient dogs to date. The use of targeted enrichment to recover genomic data from Upper Paleolithic and Mesolithic samples with low endogenous content represents an important methodological contribution, setting a benchmark for future research. The study offers valuable insights into the origins and diversification of domestic dogs, their relationships with wolves, and the demographic impacts of the Neolithic transition. These findings significantly advance our understanding of domestication processes and the long-term co-evolution of dogs and humans. Nonetheless, several aspects of the manuscript could be further refined to strengthen its overall impact.

We thank the reviewer for their kind words and useful suggestions on how to strengthen the paper.

1. For the qpAdm analyses, please provide more details on the choice of reference populations as well as the outgroups. Sensitivity analyses (e.g., varying reference sets) would strengthen the conclusions about genetic ancestry and admixture proportions.

The source and reference populations, and the way in which these are rotated across, are described in the Methods section, at the end of the "Ancestry analyses" section. We present results on sensitivity to varying source choices for the estimates of Mesolithic versus Neolithic ancestry proportions in Extended Data Figure 5, finding that the estimates are very robust to this. Additionally, we have now added two supplementary tables, one for each qpAdm analysis (1. testing for dual ancestry, 2. Mesolithic vs Neolithic ancestry contributions). These tables contain the full information for all tested models: targets, sources, reference populations, p-values, ancestry proportions and standard errors.

2. The study covers a broad time period (14–4 ka), but some regions have denser sampling than others. Acknowledge potential biases due to uneven sampling and discuss how this might affect interpretations.

We have added the following sentence to the Conclusions section to acknowledge limitations of uneven sampling when discussing the change between Paleolithic and Mesolithic European dog ancestry: "Filling the current gap in genomic sampling between 14 and 10 ka will be needed to better understand this change in European dog ancestry". We also say, when discussing the shared ancestry of pre-Neolithic dogs in Europe, that the results apply "at least in the regions of north-western Europe that our dataset covers", to caveat that our dataset does not cover southern Europe.

3. I would suggest using chromosome painting (e.g., with GLOBETROTTER) to estimate admixture timing and sources more precisely.

While we agree haplotype-based methods might be promising in principle, in practice we unfortunately don't believe they would work very well with our dataset. We are dealing with capture data that is often low coverage, from which we call pseudohaploid genotypes. Haplotype-based methods require diploid genotypes and phased haplotypes, which we don't have, and which we don't think could be very reliably inferred for this dataset. In terms of admixture timing, the primary source of information on that is the dates of the ancient samples themselves, rather than any indirect inference or extrapolation from genetic data.

4. Since ADMIXTURE is extensively described in this study, please assess stability by running multiple replicates with different random seeds.

To assess stability of ADMIXTURE to the choice of random seed, we reran ADMIXTURE (at $k=7$ as in Fig. 3a) 100 times with different random seeds, and used the CLUMPAK 'Compare' program to compare these results with our primary results shown in Fig. 3a. CLUMPAK identified a cluster of solutions that 82/100 of the replicate runs converged on, and this solution had a correlation coefficient with our primary results of 99.92%. There is thus minimal variation between runs with different random seeds, and the primary results we present in Fig 3a represents the solution most commonly found by ADMIXTURE.

We have added the above paragraph to the Methods section. Another aspect of ADMIXTURE stability is behaviour between different choices of k . To allow readers to assess this, we have also added a new figure which shows the full results for values of $k=2-10$ (new Extended Data Figure 4).

5. This study obtained time-series samples from Europe, which would be important to estimate effective population size change across time, especially pre- and post-Neolithic period.

In Figure 2C we do present results on individual heterozygosity over time, which is a function of effective population size. We find no apparent change at the time of the Neolithic transition, in contrast to what has been observed in humans. While there are more sophisticated methods for the inference of effective population size histories, such as those based on sharing of IBD (Identity By Descent) segments between individuals, they require data from a larger number of individuals than we have for any Mesolithic and Neolithic dog population, and so we need to stick to the simple and robust per-individual heterozygosity approach.

6. Even though nuclear data has higher resolution, uniparental genetic markers should also be mentioned in the main text.

In our capture data we do not target any mitochondrial markers, because the high copy number of the mitochondrial DNA can cause it to dominate the output of capture experiments. Therefore, we do not have mitochondrial data for most of the studied samples. We are also not targeting any markers on the Y chromosome, as it unfortunately is not included in the canFam3.1 reference genome. Our study does thus not advance our understanding of the uniparental history. While we agree that uniparental markers do contain useful information, it's also worth keeping in mind their limitations: mitochondrial DNA from

early dogs in Europe has been available for almost two decades, but that data did not resolve the key questions we answer here.

Some minor errors that I noticed:

1. References 8 and 20 both cite Frantz et al. (2016) with identical titles but different links.
2. Correct spelling errors (e.g., "Anotalian" to "Anatolian", line 260)
3. Check Fig. 2a, the zoomed-in PCA, whether pre-agriculture and post-agriculture are correctly labeled.

We thank the reviewer for catching these errors, and have corrected them. Regarding the pre vs post-agriculture labelling, we have also revised some of these in the process of expanding the archaeological background information on the samples (new Supplementary Material), in particular for the Danish samples, and we now describe some of these as "Uncertain" due to the uncertain cultural context at some sites.

Referee #3 (Remarks to the Author):

This is a good paper, with very interesting results about the complexity of dog ancestry and its relationship to past population movements and economic changes.

I am an archaeologist, not a geneticist. With that caveat, I do not have major concerns. My main technical comment is that I would like to see more details about the radiocarbon dating and the potential (or lack thereof) for radiocarbon reservoir offset on the directly dated specimens (see my more specific comments below).

We thank the reviewer for their kind words and useful suggestions, and respond to each point below.

Besides that, I would recommend additional contextualizing of some of the main implications of these findings. In several instances, the authors link the patterns they observe (on dogs) to known archaeological patterns about people, the most striking example being the increase in near-eastern ancestry in Europe following the Neolithic. The authors are quite terse on the implications of these findings, and what social or economic processes, intentional or not, may have been at play. What does this finding imply about social relationships between human groups, and the roles that dogs may have played?

We have deliberately avoided speculating much about the implications for the understanding of social or economic aspects of prehistoric human societies, as we believe it's very challenging to draw such conclusions confidently. Our finding that Neolithic societies incorporated Mesolithic dogs to a large extent could have many different explanations ranging from peaceful trade, to spontaneous mixing between semi-free ranging dogs, to controlled mixing, to raiding, and our genetic data does not strongly distinguish between these scenarios. This being said, on balance our results could be seen as speaking against a very violent and sudden process of replacement, similar to what happened in post-Columbian Americas. We have added the following sentence to the final paragraph of the Conclusions, to briefly allude to this:

“Our results suggest a less sudden and violent process in Neolithic Europe, consistent with admixture and interactions between farmers and hunter-gatherers ongoing for several centuries in many parts of the continent (Lipson et al. 2017). The persistence of Mesolithic dog ancestry in the Neolithic might imply that hunter-gatherers and farmers used dogs for at least partially overlapping purposes, but we stress that genetic data can not reveal the social and economic processes underlying the admixture between Mesolithic and Neolithic dogs.

Additionally, we have added/expanded some additional contextualisation in the following two instances.

- When discussing the Neolithic turnover in Denmark: “Both dogs from the Neolithic-Mesolithic transitional site of Syltholm were consistent with having little or no Neolithic Southwest Asian ancestry, mirroring human DNA from chewed birch pitch at the site showing completely Mesolithic ancestry (Jensen et al. 2019)”
- When discussing the initial spread of dogs through Europe and how it relates to the human history: “Coincidentally, the earliest observation of the Villabruna ancestry north of the Alps is in individuals from the Bonn-Oberkassel site in Germany 14 ka (Posth et al. 2023), where they were buried alongside one of the earliest known canids with clear dog morphology (Janssens et al. 2018) (though nuclear genetic data is not available from this likely dog)”

I would also suggest being more explicit about some reasoning shortcuts. I list a few below; the authors for instance implicitly assume that all ancient dogs must be genetically related to modern dogs; or that a sampled bone sort of equates an ancient individual.

I list some more comments below.

We thank the reviewer for these suggestions, and have tried to be more explicit about our reasoning in these contexts, as further described below.

line 106. indicate that (presumably) all dates in this paper are calibrated

All radiocarbon dates have been calibrated, with both uncalibrated and calibrated dates provided in Supplementary Data 2. This specific sentence in line 106, “from the period 14-17 ka.”, does not refer to a radiocarbon date but just generally to a time period in the past, such that the question of calibration does not apply.

line 108: reference 17 for the Americas seems a bit outdated – surely there is more recent work to be cited here, if nothing else by some of the authors themselves

We agree, and have changed this reference to the following:

Perri, A., Widga, C., Lawler, D., Martin, T., Loebel, T., Farnsworth, K., Kohn, L., & Buenger, B. (2019). NEW EVIDENCE OF THE EARLIEST DOMESTIC DOGS IN THE AMERICAS. *American Antiquity*, 84(1), 68–87.

lines 135-144. It would be good to explain how samples were selected. Do they represent different individuals, and if so how did the authors make sure they do? I imagine for instance that the 103 samples from Kesslerloch do not represent a MNI of 103. If they represent the same individuals, how does it affect the results?

We agree that this aspect is potentially confusing, and we have added the following to the beginning of the Methods section on sampling to hopefully clarify:

“When sampling skeletal remains for DNA, in most instances we aimed to take one sample per biological individual. In some instances, samples were obtained from different remains that possibly could derive from the same individual, based on the zooarchaeological context (in particular this was the case for the Kesslerloch site). In a few cases where it was unambiguous that data from multiple remains came from the same individual, we merged those data. But in most cases, we treated data from different remains separately, to err on the side of caution and not incorrectly merge data. The true number of sampled individuals is thus unknown, but we provide a table listing the groups of remains where archaeological information suggest they might derive from the same individual (Supplementary Data 5).”

Furthermore, we have altered our phrasing in the main text to refer to “remains” rather than “samples” in applicable places, e.g. when talking about the number of remains included in the study. To further clarify this to the reader, we have also added this sentence to the main text paragraph introducing the samples: “Some of these remains might derive from the same biological individual.”

This is an issue mainly just for the Kesslerloch site, but here it only concerns wolf individuals, the data from which we do not analyse further in this paper beyond concluding that they are wolves. The number of unique individuals represented among those remains thus has no implications for our results or conclusions. For the key Kesslerloch individual that is a dog, all data derives from a single element (a maxilla).

One possibility would be to try to infer from the genetic data what remains correspond to the same individual, but we have not attempted this given that it's not information that we need in order to answer our research questions. Given the variable amounts of data obtained, it would likely work for some pairs of remains, but not others. The data is available for any other researchers who would like to attempt this inference.

lines 171-180. the authors start by discussing how their test separates “known dogs” from wolves, then switch to how it separates [implicitly all] “dogs” from wolves. Perhaps they ought to discuss how they make this leap, or at least explicitly state it. Is it inconceivable that there could be ghost dog lineages unrelated to “known dogs”? If it is so, could the authors explain why?

It is conceivable that there could have been other, independently domesticated wolf populations, or “ghost dogs”, which if so would not be detected in our analyses - we acknowledge and briefly discuss this in the first paragraph of the next section: “If they shared no ancestry with other dogs, they would look no different from wolves in these analyses.”

If such a population existed, it would probably be wise to call them something other than “dogs”, to avoid confusion with *Canis lupus familiaris*. Our paper is only concerned with dogs as we know them, that is members of the *Canis lupus familiaris* lineage, and we do not intend for the word ‘dog’ to refer to any domesticated wolf or canid.

While ghost domestication events can never be ruled out, we also think there is no particular reason to constantly caveat our writing to account for their possibility. One could apply the same reasoning to any animal studied through ancient DNA. E.g. if we sequence the genome of a mammoth, we can't strictly rule out that it was part of some unknown ghost population of domesticated mammoths. But in the absence of any specific evidence that there might have been ghost domesticates, we think it's reasonable for the term 'dog' to simply refer to dogs as we know them. We have added the following sentence to the paragraph discussing the test for distinguishing dogs and wolves, to clarify our reasoning on this point:

"Strictly speaking, genetic data cannot rule out that any of the individuals we classify as wolves had been domesticated independently and thus could also be considered 'dogs' in some sense, but in this manuscript we only intend the word 'dog' to refer to members of the known *Canis lupus familiaris* population."

lines 178-188. can the authors indicate the time frame for all sites? as it stands, it is only given for a select few. This comment applies to the entire article.

The time frames for the sets of sites are given in the first paragraph of the previous section (Targeted genome-wide DNA enrichment), but we see the reviewer's point that the reader might not necessarily remember these. To enable easy reference to sites and their time frames, rather than repeating these multiple times in the main text, we have added this information to the sample map in Figure 1a.

line 187-188. the statement on hunting being the most parsimonious explanation comes a bit out of nowhere. I would suggest either expanding on this argument or removing it.

This is not an argument that we make based on any of our specific findings, rather we refer to zooarchaeological literature that describes widespread hunting of wolves and other carnivores in pre-Neolithic Europe. To make this clearer, we have moved the citation to the end of the sentence, such that it also covers the hunting explanation. Another possible explanation for why we find wolves at Mesolithic sites could have been that humans had some kind of special relationship to wolves, perhaps taming them, but we see no reason to propose something spectacular like that when hunting can explain our findings in a more parsimonious way.

line 192. does this finding contradict the argument above that wolves were obtained by hunting?

This sentence concerns a single wolf, with traces of human modification and red staining. We can't know how this individual lived or died, but in any case this single observation does not substantially change the overall conclusion from the zooarchaeological literature that hunting is likely the most common explanation for why wolves are found at pre-Neolithic human sites. It's also still very much possible that this particular individual was indeed hunted (the modification marks suggest it was eaten by humans).

line 194: other ways such as?

We have added “(e.g. taming)” here to indicate that this is not a type of interaction that can be ruled out by the genetic data.

lines 198-201. are the bones found in stratigraphic contexts that suggest a late deposition?

We don't have any prior evidence for later deposition of these dogs with recent ancestry, which made the results even more surprising. Some of them come from early excavations, where less care might have been taken to record stratigraphic contexts. In any case, with radiocarbon and/or genetic data we can clearly determine that these dogs are not of Late Pleistocene age.

line 202-204. likewise, can the authors explain this major dating discrepancy?

We don't really have an explanation, but it must reflect later deposition: a dog that was found in a seemingly Neolithic context turned out to be Late Iron Age. We don't have anything further useful to add on this point, but it's good that we can correct the date for this previously published dog genome.

line 206-207. this partially answers my query for lines 135-144, but only partially. does this means Kesslerloch has 62 individuals instead of 103? What of the other sites?

The number 62 is the number of samples that we found to have wolf ancestry (“we find that 62 have wolf ancestry and one is from a dog”), and so is a separate matter from whether any of the samples come from the same biological individual.

line 207. is the dog from the same archaeological context than the wolves? it seems odd to be by itself among so many wolves.

The dog is from the same archaeological context as the many wolves, yes. We agree it is surprising that there is just a single dog amongst all of these wolves, but this is what we found. This is also in agreement with the prior morphological analyses of the Kesslerloch material, which did not identify any other potential dogs in the assemblage. We have added a description of the Kesslerloch site and excavation to the Supplementary material.

lines 170-211. Is there any dietary information on these dogs, such as stable isotopes? Given the likelihood that at least some of these people (and their dogs) may have eaten aquatic resources, the authors should discuss how radiocarbon reservoir effect may affect their dates (or not).

We return to the isotope question in the very last response. But briefly, we do not have indications that any of the radiocarbon dates obtained in this study would require marine reservoir effect correction, and have added a sentence to the Methods to clarify this.

lines 343-346. can the authors relate this SW Asia / Magdalenian canid connection with connections (or lack thereof) observed among human populations at that time?

As noted in the reviewer comment after next, we get to comparisons to the human population history in the Conclusions section.

line 412-414. this statement makes it sound as if “Mesolithic people” were replaced by “Neolithic people” and went extinct. Many of these farmer groups probably included people (and their dogs) from Mesolithic backgrounds.

We have rephrased this sentence to focus on the dogs rather than the people, and say: “This suggests that dogs from local hunter-gatherer groups contributed substantially to the dog populations that lived with Neolithic farmer communities in Europe.”

We have also rephrased a similar sentence in the abstract, to say: “suggesting that dogs from Mesolithic hunter-gatherer groups contributed substantially to Neolithic dog populations in Europe, and ultimately likely also to modern European dogs”

line 445-446. I see that this responds to my comment in lines 343-346. It could be good to mention that earlier.

We feel like the Conclusions section is the most natural place to reflect on possible similarities to human population history. We have also expanded this slightly by adding the following, which represents a quite noteworthy observation:

“Coincidentally, the earliest observation of the Villabruna ancestry north of the Alps is in individuals from the Bonn-Oberkassel site in Germany 14 ka (Posth et al. 2023), where they were buried alongside one of the earliest known canids with clear dog morphology (Janssens et al. 2018) (though nuclear genetic data is not available from this likely dog). A tempting hypothesis is thus that dogs spread through Europe with, or just ahead of, the Villabruna expansion.”

lines 455-456. here (or somewhere else) the authors could cite the new paper by Manin et al. that discusses how dog populations were affected by the “Neolithic transition” in South America.

We thank the reviewer for this suggestion, but on consideration do not think South America represents a comparable history. Manin et al. 2025 favours a model in which hunter-gatherers in South America did not have dogs at all, and that dogs only arrived in South America with the expansion of maize agriculturalists. Therefore, this would not represent a comparable data point on what happens when human cultures expand into regions where there already are local dogs.

lines 455-456. It could also be good to discuss the differing roles that dogs may have played among hunter-gatherers vs farmers populations.

This is a very interesting question, though we don’t feel very comfortable speculating too much about this, as unfortunately we just don’t know very much about those roles. This being said, one possible take-away could be that, since Mesolithic dogs contributed substantially to Neolithic dog populations, that these societies would have used dogs for at least partially overlapping purposes. If Mesolithic dogs were useless to Neolithic societies because they fulfilled the wrong roles, they likely would not have been incorporated. We now

allude to this in the sentences we have added in the final paragraph of the Conclusions section:

“Our results suggest a less sudden and violent process in Neolithic Europe, consistent with admixture and interactions between farmers and hunter-gatherers ongoing for several centuries in many parts of the continent [Lipson et al. 2017]. The persistence of Mesolithic dog ancestry in the Neolithic might imply that hunter-gatherers and farmers used dogs for at least partially overlapping purposes, but we stress that genetic data can not reveal the social and economic processes underlying the admixture between Mesolithic and Neolithic dogs.”

lines 582-584. As mentioned in a previous comment, it would be good to explain the discrepancy here.

As mentioned in reply to the previous comment, we don't have any useful explanation for why the contextual attribution of this previously published dog was incorrect.

lines 579-584. As mentioned earlier, the authors should discuss the impact (or lack thereof) of diet on potential radiocarbon reservoir effect.

We elaborate on $\delta^{13}\text{C}$ values in the response to the next comment. Briefly, we don't obtain any $\delta^{13}\text{C}$ values that suggest marine diets, and so we do not believe that any marine reservoir correction is necessary. We have added the following sentence to the methods section describing the radiocarbon dates:

“For samples where $\delta^{13}\text{C}$ values were obtained separately from the AMS measurement by high-precision stable isotope mass spectrometry, and thereby inform on diet, no high values indicative of potential marine diets were observed, and thus no corrections for marine reservoir effect was applied.”

Supplemental excel file. The large variation in $\delta^{13}\text{C}$ from -14 to -24 makes me wonder, again, how diet may affect radiocarbon dates (particularly the values above -14). Are those values expected of canids with fully terrestrial diets for these areas? Are $\delta^{15}\text{N}$ available? Are isotopic values ($\delta^{13}\text{C}$ and $\delta^{15}\text{N}$) available for the dated Kesslerloch specimens?

We thank the reviewer for pointing this out, and have taken the opportunity to clarify this in the table reporting new radiocarbon dates (Supplementary Data 2). Some of the $\delta^{13}\text{C}$ values reported were obtained directly from the AMS radiocarbon dating, where they are used to correct for isotopic fractionation, and are not from a stable isotope mass spectrometer experiment (e.g. IRMS). $\delta^{13}\text{C}$ values from AMS experiments are less accurate and not comparable to those from IRMS, and as such should not be interpreted in terms of diet. We have clarified this in the table by splitting the column into two, “ $\delta^{13}\text{C}$ (AMS)” and “ $\delta^{13}\text{C}$ (Stable isotope mass spectrometer)”. This shows that the very high values that the reviewer noticed (e.g. -14.2 and -16) are from AMS experiments, and thus should not be interpreted in terms of diet. For the samples where we do have high-precision stable isotope measurements of $\delta^{13}\text{C}$, values are typical of terrestrial diets (highest value is -20.05) and do not imply that marine reservoir correction should be necessary.

In the spirit of further increasing clarity and transparency for the new radiocarbon dates, we have also added a column on “Pretreatment chemistry”, adding this information for those samples where we have been provided with it from the radiocarbon dating facility, in line with suggestions from a recent paper on improving standards for radiocarbon date reporting (<https://doi.org/10.1002/jqs.70012>).

On the topic of $\delta^{15}\text{N}$ values, this is not something that we have generated in the current study, where the focus is genetic ancestry rather than diet. Some stable isotope data has previously been published on some of the specimens analysed here, including some Kesslerloch specimens. Bocherens et al. 2011 (<https://doi.org/10.1016/j.quaint.2011.02.023>) and Baumann et al. 2020 (<https://doi.org/10.1016/j.quascirev.2019.106032>) reported data from a total of five Kesslerloch canids, with the $\delta^{13}\text{C}$ values falling between -20.1 and -19.0, and $\delta^{15}\text{N}$ values of between 5.7 and 7.9. The latter study included the individual identified here as a dog genetically (M001:189), implying it has a similar diet to the wolves at the site. As with the radiocarbon dates newly generated for this study, these isotope data give no reason to think that marine reservoir correction would be necessary for the previously published radiocarbon date for the Kesslerloch dog.

François Lanoë

We have made the following changes to the text in response to reviewer comments:

- Changed the abstract following referee #2's first suggestion, from "two-thirds of the tested remains" to "141 out of 216 remains"
- Added the following sentence to the final Conclusions paragraph, following referee #2's second suggestion: "The smaller Neolithic impact on dogs is also reflected in how they do not mirror the genetic diversity increase seen in humans."

We have made the following changes in response to editor requests:

- Shortened the abstract from 288 to 230 words.
- Shortened all subheadings to 40 characters or less (one is at 41 characters)
- We have added "on a base map from Natural Earth (naturalearthdata.com)" to the figure legend for Fig 1a, in response to query 8 about third party figures. Maps from Natural Earth can be used freely in any manner (<https://www.naturalearthdata.com/about/terms-of-use/>). We have no other third party materials, and so are not submitting the "THIRD PARTY MATERIAL INFORMATION" form.
- We have added a competing interests statement.
- We have made the ENA study public.